# TICO-19: the Translation Initiative for COvid-19

**Antonios Anastasopoulos**[j]**, Alessandro Cattelan**[ʏ]**, Zi-Yi Dou**[ʎ]**, Marcello Federico**[ς]**,**
**Christian Federman**[ð]**, Dmitriy Genzel**[ℙ]**, Francisco Guzmán**[ℙ]**, Junjie Hu**[ʎ]**, Macduff Hughes**[ᴈ]**,**
**Philipp Koehn**[ꝛ]**, Rosie Lazar**[ϕ]**, Will Lewis**[ð]**, Graham Neubig**[ʎ]**, Mengmeng Niu**[ᴈ]**,**
**Alp Öktem**[ꭗ]**, Eric Paquin**[ꭗ]**, Grace Tang**[ꭗ]**, Sylwia Tur**[ϕ]

[j]Department of Computer Science, George Mason University
[ʎ]Language Technologies Institute, Carnegie Mellon University
[ʏ]Translated [ς]Amazon AI [ð]Microsoft [ℙ]Facebook [ꝛ]Johns Hopkins University
[ϕ]Appen [ᴈ]Google [ꭗ]Translators without Borders
tico19.2020@gmail.com

## Abstract

The COVID-19 pandemic is the worst pandemic to strike the world in over a century. Crucial to stemming the tide of the SARS-CoV-2 virus is communicating to vulnerable populations the means by which they can protect themselves. To this end, the collaborators forming the **T**ranslation **I**nitiative for **CO**vid-**19** (TICO-19)[1] have made test and development data available to AI and MT researchers in 35 different languages in order to foster the development of tools and resources for improving access to information about COVID-19 in these languages. In addition to 9 high-resourced, "pivot" languages, the team is targeting 26 lesser resourced languages, in particular languages of Africa, South Asia and South-East Asia, whose populations may be the most vulnerable to the spread of the virus. The same data is translated into all of the languages represented, meaning that testing or development can be done for any pairing of languages in the set. Further, the team is converting the test and development data into translation memories (TMXs) that can be used by localizers from and to any of the languages.[2]

## 1 Introduction

The COVID-19 pandemic marks the worst pandemic to strike the world since 1918. At the time of this writing,[3] the SARS-CoV-2 coronavirus responsible for COVID-19 has infected over ten million people worldwide, with over a half a million deaths. While these numbers are likely under-reported, they are growing at an alarming rate, and many millions of people could become infected or perish without proper prevention measures.

Effective communication from health authorities is essential to protect at-risk populations, slow down the spread of the disease, and decrease its morbidity and mortality (UNOCHA, 2020). Yet, preventive measures such as stay-at-home orders, social distancing, and requirements to wear personal protective equipment (e.g. masks, gloves, etc.) have proven difficult to relay. That's not accounting for the difficulty to disseminate correct technical information about the disease, such as symptoms (e.g., fever, chills, etc.), specifics about testing (e.g., viral *vs.* antibody testing), and treatments (e.g., intubation, plasma transfusion).

While official communications from the World's Health Organization (WHO) are constantly published and revised, they are mostly limited to major languages. This has resulted in a vacuum in many languages that has been filled by an *infodemic* of misinformation, as described by the WHO. Non-governmental organizations (NGOs) such as Translators without Borders (TWB) play an important role in delivering multilingual communication in emergencies, such as the COVID-19 pandemic, but their reach and capacity has been outsized by the needs presented by the pandemic. To date, TWB has translated over 3.5 million words with over 80 non-profit organizations for more than 100 language pairs as part of their COVID-19 response.

Translation technologies such as automatic Machine Translation (MT) and Computer Assisted Translation (CAT) present unique opportunities to scale the throughput of human translators. However, given the sensitivity of the content, it is critical that the translations produced automatically are of the highest possible quality.

The **T**ranslation **I**nitiative for **CO**vid-**19** (TICO-19) effort marks a unique collaboration between

---

[1]Collaborators in the initiative include Translators without Borders, Carnegie Mellon University, Johns Hopkins University, George Mason University, Amazon Web Services, Appen, Facebook, Google, Microsoft, and Translated.

[2]The dataset, translation memories, and additional resources are freely available online: http://tico-19.github.io/. As the project continues and we create data for more languages, we will keep updating this paper as well as the project's website.

[3]July 1st, 2020

public and private entities that came together shortly after the beginning of the pandemic.[4] The focus of TICO-19 is to enable the translation of content related to COVID-19 into a wide range of languages. First, we make available a collection of translation memories and technical glossaries so that language service providers (LSPs), translators and volunteers can make use of them to expedite their work and ensure consistency and accuracy. Second, we provide an open-source, multi-lingual benchmark set (which includes data for very-low-resource languages) specialized in the medical domain, which is intended to track the quality of current machine translation systems, thus enabling future research in the area. Lastly, we provide mono-lingual and bi-lingual resources for MT practitioners to use in order to advance the state-of-the-art in medical and humanitarian Machine Translation, as well as other natural language processing (NLP) applications.

Our hope is that our work will in the short-term enable the translation of important communications into multiple languages, and that in the long-term, it will serve to foster the research on MT for specialized content into low-resource languages. Through these resources we hope that our society is better prepared to quickly respond to the needs of translation in the midst of crises (e.g., for future crises, *a la* Lewis et al. (2011)).

## 2 The Value of Translation Technologies in Crisis Scenarios

During a crisis, whether it is local to one region or is a worldwide pandemic, communicating effectively in the languages and formats people understand is central to effective programs on the ground. For example, as part of the effort to control the spread of COVID-19, the Global Humanitarian Response Plan recognizes community engagement in relevant languages as a key strategy (UNOCHA, 2020).[5] In some countries, this will be all the more

vital because information will be the main defense against the disease, and particular effort will be needed to make it accessible and grounded in local culture and context. Among these are countries where large sections of the population do not speak the dominant language.

Historically, MT, NLP and translation technologies have played a crucial role in crisis scenarios. The response to the Haitian earthquake in 2010 was notable for the broad use of technology in the humanitarian response, relying more on crowd-sourced translations and geolocation, but notably, translation technology was also used. In the days following the earthquake, Haitian citizens were encouraged to text messages requesting assistance to "4636", and as many as 5,000 messages were texted to this number per hour. Unfortunately for the aid agencies, whose dominant languages were English and French (aid agencies included the US Navy, the Red Cross, and Doctors without Borders), most of the SMS messages were in Haitian Kreyòl. Quickly, the Haitian Kreyòl speaking diaspora around the world were activated by the Mission 4636 consortium to translate the SMS messages and geolocate (Munro, 2010), and the translated messages were handed off to aid agencies for triage and action. The Mission 4636 infrastructure included a high-precision rule-based MT (Lewis et al., 2011), and within days to weeks after the earthquake, statistical MT engines were brought online by Microsoft (Lewis, 2010) and Google.[6]

Translation technology continues to be used in a variety of crisis and relief scenarios. Notable among these is Translators without Borders (TWB) use of translation memories for translating to a number of under-resourced languages in relief settings. Likewise, the Standby Task Force,[7] who are activated in a variety of relief settings, note the use of MT in various deployments around the world, e.g., for Urdu in the Pakistan earthquake of 2011 and for Spanish in the Ecuador earthquake in 2016. The EU funded *INTERnAtional network on Crisis Translation* (INTERACT)[8] project, started

---

[4] The World Health Organization (WHO) declared COVID-19 a pandemic on March 11th, 2020. The TICO-19 collaborators came together in the days following and first met as a group (over Zoom) on March 20th. It cannot be understated the rapidity with which this collaboration came together and how seamlessly the participants, many erstwhile competitors, have worked in harmony and without animosity. It is truly a testament to the needs of the greater good outweighing personal differences or potentially conflicting objectives.

[5] The plan does not call out Machine Translation *per se*, but does call out the need for content to be produced and disseminated in "accessible languages", and the need for communication in "local languages".

[6] Although these engines were not integrated into the relief pipeline developed by Munro and colleagues, Lewis et al. (2011) document how MT *could* be integrated into a crowd-centric relief pipeline like that used by Mission 4636, whereby MT, even if low-precision or noisy, could provide first-pass translations which could then be triaged before being handed off to translators for more accurate translations and geolocation.

[7] https://www.standbytaskforce.org/
[8] https://cordis.europa.eu/project/id/734211

a couple of years before the COVID-19 pandemic, focused on crisis translation, specifically in health crises such as pandemics, with a focus on improving resilience in times of crises through communication, ultimately with the goal of reducing loss of life.[9] Likewise, during the current pandemic, several community-driven efforts have sprung up to fulfil the need for information communication. The Endangered Languages Project[10], for example, has collected community-produced translations of public health information in hundreds of languages in various formats.

What is not tracked is the degree to which publicly available MT tools and resources are used in crisis and relief settings, e.g., translation apps and tools from Amazon, Google and Microsoft, or the translation feature built into Facebook (e.g., automatically translating posts). The authors suspect use may be broad, but there are no published accounts documenting just how broadly and how much these tools are used in crises. Tantalizing evidence of the use of publicly available tools was noted by Lewis et al. (2011) who documented traffic in the Microsoft Translator apps in the weeks following the Haitian earthquake: they noted that at least 5 percent of the Haitian Kreyòl traffic was relief-related. It is likely that Google's and Microsoft's apps are used even when cell phone infrastructure is unavailable or destroyed, since the tools permit users to download models to their devices so they can perform offline translations.[11]

In crises, it is clear that organizations need the capacity to communicate critical information and key messages into the languages people understand, at speed and at scale. Crisis affected communities could access content in local languages through various channels such as SMS, online chatbots, or more traditional printed materials. Their questions and feedback can be used to refine content to better meet their needs. Likewise, relief agencies need access to SMS and other communiques in local languages in order to more effectively and equitably distribute aid.

MT can help the various actors to translate and disseminate essential communications in a timely manner without the need to wait on human transla-

tors. This is particularly important in low resourced languages where professional translators are not readily available. Domain-specific MT can also assist translators with the right terminology to convey the correct response and standardize concepts. Furthermore, people who are unable to understand major languages could get access to vital information (such as news sources, websites, etc) first hand via a MT-driven tool set. We emphatically note that communication is not *just* a translation problem. Especially in the case of under-represented indigenous communities, messaging needs to also remain respectful of cultural norms (e.g. communicate through appropriate channels, without undermining cultural authorities and practices) and to not minimize the agency of such communities through "deficit framing" (Wanambi et al., 2020). Nevertheless, translation is a crucial component of the information flow.

However, to be useful for translating specialized content such as medical texts, we require that automatic translations be of the highest possible accuracy. To advance the research in Machine Translation, we require both high-quality benchmark sets and access to basic training resources, both monolingual and parallel. Likewise, translation memories in a broader set of languages can help localizers around the world translate into these languages. In the remainder of this paper we describe the resources created by the TICO-19 initiative, and some evaluations against them.

## 3 The TICO-19 Translation Benchmark

We created the TICO-19 benchmark with three criteria in mind: diversity, relevance and quality. First, we sampled from a variety of public sources containing COVID-19 related content, representing different domains. Second, to make our content relevant for relief organizations, we chose the languages to translate into based on the requests from relief organizations *on-the-ground*. Third, we established a stringent quality assurance process, to ensure that the content is translated according to the highest industry standard.

### 3.1 COVID-19 source data

The translation benchmark was created by combining English open-source data from various sources, listed in Table 1. We took special care to diversify the domains and sources of the data. We provide a concise summary here and detailed statistics for

---

[9]The project has already released COVID-19-specific MT models for 4 languages (Way et al., 2020).

[10]https://endangeredlanguagesproject.github.io/COVID-19/.

[11]Crucially, it should be noted that these apps and tools are available for no more than 110 languages, leaving most of the world's languages in the dark.

| Data Source | Domain | Statistics | | | |
|---|---|---|---|---|---|
| | | #docs | #sents | #words | avg. slen |
| CMU | medical, conversational | – | 141 | 1.2k | 8.5 |
| PubMed | medical, scientific | 6 | 939 | 21.2k | 22.5 |
| Wikinews | news | 6 | 88 | 1.8k | 20.4 |
| Wikivoyage | travel | 1 | 243 | 4.5k | 18.5 |
| Wikipedia | general | 15 | 1,538 | 38.1k | 24.7 |
| Wikisource | announcements | 2 | 122 | 2.4k | 19.6 |
| **Total** | | **30** | **3,071** | **69.7k** | **22.7** |
| *Dev Set* | | 12 | 971 | 21.0k | 21.6 |
| *Test Set* | | 18 | 2,100 | 49.3k | 23.5 |

Table 1: Source-side (English) statistics of the TICO-19 benchmark.

every source in Appendix B:

- **PubMed**: we selected 6 COVID-19-related scientific articles from PubMed[12] for a total of 939 sentences.

- **CMU English-Haitian Creole dataset** (CMU): the data were originally collected at Carnegie Mellon University[13] and translated into Haitian Creole by Eriksen Translations, Inc. The dataset is comprised of medical domain phrases and sentences, which along other data were used to quickly build and deploy statistical MT systems in disaster-ridden Haiti (Lewis, 2010) and later was part of the 2011 Workshop of (Statistical) Machine Translation Shared Tasks (Callison-Burch et al., 2011). For our purposes, we sub-sampled the English conversational phrases to only those including COVID-19-related keywords taken from our terminologies (see Section 4), ending up with 140 sentences.

- **Wikipedia**: we selected 15 COVID-19-related articles from the English Wikipedia[14] on topics ranging from responses to the pandemic, drug development, testing, and coronaviruses in general.

- **Wikinews, Wikivoyage, Wikisource**: 6 COVID-19-related entries from Wikinews.[15] one article from Wikivoyage[16] summarizing travel restrictions, and two entries from Wikisource[17] (an executive order and an internal Wikipedia communiqué). These data respectively cover the domains of news, travel advisories, and government/organization announcements.

### 3.2 Languages

We translated the above English data into 38 languages.[18] In some cases, this was achieved through pivot languages, i.e., the content was translated into the pivot language first (e.g., French, Farsi) and then translated into the target language (e.g., Congolese Swahili, Dari). The languages were selected according to various criteria, with the main consideration being the potential impact of our collected translations and the humanitarian priorities of TWB. The translation languages include:

- **Pivots:** 9 major languages which function as a *lingua franca* for large parts of the globe: Arabic (modern standard), Chinese (simplified), French, Brazilian Portuguese, Latin American Spanish, Hindi, Russian, Swahili, and Indonesian.
- **Priority:** 21 languages which TWB classified as high-priority, due to the large volume of requests they are receiving and the strategic location of their partners (e.g. the Red Cross). They include languages in Asia –Dari, Central Khmer, Kurdish Kurmanji (Latin script), Kurdish Sorani (Arabic script), Nepali, Pashto– and Africa –Amharic, Congolese Swahili,

[12]https://www.ncbi.nlm.nih.gov/pubmed/
[13]Under the NSF-funded (jointly with the EU) "NE-SPOLE!" project.
[14]https://en.wikipedia.org
[15]https://en.wikinews.org
[16]https://en.wikivoyage.org
[17]https://en.wikisource.org
[18]All translations are available under a CC0 license.

| Data Source | Example |
|---|---|
| CMU | are you having any shortness of breath? |
| PubMed | The basic reproductive number (R0) was 3.77 (95% CI: 3.51-4.05), and the adjusted R0 was 2.23-4.82. |
| Wikinews | By yesterday, the World Health Organization reported 1,051,635 confirmed cases, including 79,332 cases in the twenty four hours preceding 10 a.m. Central European Time (0800 UTC) on April 4. |
| Wikivoyage | Due to the spread of the disease, you are advised not to travel unless necessary, to avoid being infected, quarantined, or stranded by changing restrictions and cancelled flights. |
| Wikipedia | Drug development is the process of bringing a new infectious disease vaccine or therapeutic drug to the market once a lead compound has been identified through the process of drug discovery. |
| Wikisource | The federal government has identified 16 critical infrastructure sectors whose assets, systems, and networks, whether physical or virtual, are considered so vital to the United States that their incapacitation or destruction would have a debilitating effect on security, economic security, public health or safety, or any combination thereof. |

Table 2: Samples of the English source sentences for the TICO-19 benchmark.

Dinka, Nigerian Fulfulde, Hausa, Kanuri, Kinyarwanda, Lingala, Luganda, Nuer, Oromo, Somali, Eritrean Tigrinya, Ethiopian Tigrinya, Zulu.

- **Important:** 8 additional languages spoken by millions in South and South-East Asia: Bengali, Burmese (Myanmar), Farsi, Malay, Marathi, Tagalog, Tamil, and Urdu.

The latter two sets are primarily languages of Africa, and South and South-East Asia, whose communities, according to on-the-ground organizations, may be most susceptible to the spread of the virus and its potentially disastrous ramifications, mostly due to lack of access to information and communication in the community languages. They are also overwhelmingly under-resourced languages; in fact, some of the languages have remained untouched by the AI and MT communities, and have no known tools or resources that have been developed for them.

All of the test and development documents are sentence aligned across all of the languages, which allows for any pairing of languages for testing or development purposes. This was done by design, in order to facilitate tool and resource development in and across any of the targeted languages. For example, an MT developer could develop translation sys-

tems for French to/from Congolese Swahili, Arabic to/from Kurdish, Urdu to/from Pashto, Hindi to/from Marathi, Amharic to/from Oromo, or Chinese to/from Malay, among the 1296 possible pairings. Note that as the project continues and as we create data for more languages we will keep updating this paper as well as the project's website.

### 3.3 Quality Assurance

It has been observed that translation from and into low-resource languages requires additional automatic and manual quality checks (Guzmán et al., 2019). To obtain the highest possible quality, here we implemented a two-step human quality control process. First, each document is sent for translation to language service providers (LSP), where the translation is performed. After translation, the dataset goes through a process of *editing*, in which each sentence is thoroughly vetted by qualified professionals familiar with the medical domain, whenever available.[19] In case of discrepancies, a process of arbitration is followed to solve disagreements between translators and editors.

After editing, a selected fraction of the data (18%, 558 sentences) undergoes a second inde-

---

[19]For a handful of languages, such as Dinka or Nuer, where simply creating translations was a challenge, this process was, by necessity, skipped.

pendent quality assurance process. To ensure quality in the hardest-to-translate data, the scientific medical content from PubMed was upsampled so that it comprises 329 of the 558 doubly-checked sentences (almost 59%). The exact documents that comprise our second quality assurance set are listed in Appendix C.

The quality of the translations was checked, and reworks were made until every translation set was rated above 95% across all languages, before any additional subsequent edits. Some low-resource languages like Somali, Dari, Khmer, Amharic, Tamil, Farsi, and Marathi required several rounds of translation to reach acceptable performance. The hardest part, unsurprisingly, proved to be the PubMed portion of the benchmark. Our QA process revealed that in most cases the problems arose when the translators did not have any medical expertise, which lead them to misunderstand the English source sentence and often opt for sub-par literal or word-for-word translations. We provide additional details with the estimated quality per language in the Appendix D. We note that all mistakes identified in this subset have been corrected in the final released dataset, and that all sentences that underwent the QA process are part of the test portion of our benchmark.

We additionally release the sampled dataset along with detailed error annotations and corrections. Whenever an error was noted in the validation sample, it was classified as one of the following categories: Addition/Omission, Grammar, Punctuation, Spelling, Capitalization, Mis-translation, Unnatural translation, and Untranslated text. The severity of the error was also classified as *minor*, *major*, or *critical*. Although small in size (at most 558 sentences in each translation direction), we hope that releasing these annotations will also invite automatic quality estimation and post-editing research for diverse under-resourced languages.

## 4 Translator Resources

**Translation Memories**   Because of the breadth of languages covered by TICO-19, and the fact that so many are under-resourced, the translations themselves can be of significant value to localizers. As part of the effort, the TICO-19 collaborators have converted[20] the translated data to translation memories, cast as TMX files, for all

English–X pairings, as well as some other pairings of languages focusing on potential local needs (e.g. French–Congolese Swahili, Farsi–Dari, and Kurdish Kurmanji–Sorani). These TMX files, in addition to the test and development data, have been made available to the public through the project's website.

**Terminologies**   Two sets of translation terminologies were provided by Facebook and Google (the complete set of the English source terms and of the translated languages are listed in Appendix E):

- the **Facebook** one includes 364 COVID-19 related terms translated in 92 languages/locales.

- the **Google** one includes 300 COVID-19 related terms translated from English to 100 languages and a total of 1300 terms from 27 languages translated into English (for a total of approximately 30k terms).

**Additional Translations**   Translators without Borders (TWB) worked with its network of translators to provide translations in hard-to-source languages (e.g. Congolese Swahili and Kanuri). It also provided COVID-19 specific sources from its diverse humanitarian partners to augment the dataset. This augmented dataset will be available under license on TWB's Gamayun portal.[21]

## 5 MT Developer Resources

As part of our project the CMU team also collected some COVID-19-related monolingual data in multiple languages. They are available online,[22] but we note that some of these data might not be available under the same license as our datasets (and hence might not be appropriate for commercial system development). These are detailed in the next sections.

### 5.1 Monolingual

**Wikipedia Data**   COVID-19-related data from Wikipedia were scraped in 37 languages. COVID-19 terms were used as queries (language specific, in most cases), retrieving the textual data from the returned articles (i.e. stripping out any Wikipedia markup, metadata, images, etc). The data ranges from less than 1K sentences (around 10K tokens) for languages like Hindi, Bengali, or Afrikaans, and for as much as 8K sentences for Spanish (160K tokens) or Hungarian (120K tokens).

---

[20]Conversion was carried out with the Tikal command included in the Okapi framework: https://okapiframework.org/.

[21]https://gamayun.translatorswb.org/
[22]https://bit.ly/2ZLOkpo

**News Data**  COVID-19-related news articles (as identified by keyword search) were scraped from three news organizations that publish multilingually through their world services. Specifically, the collected data include articles from the BBC World Service[23] (22 languages), the Voice of America[24] (31 languages), and the Deutsche Welle[25] (29 languages).

## 5.2  Parallel

We have also scraped a very small amount of available parallel data, mostly from public service announcements from NGOs and national/state government sources. Specifically, we scraped Public Service Announcements by the Canadian government[26] in 21 languages (English, French, and First Nations Languages), a fact sheet provided by the King County (Washington, USA)[27] in 12 languages, a COVID-19 advice sheet from the Doctors of the World[28] in 47 languages, and data from the COVID-19 Myth Busters in World Languages project[29] in 28 languages, and a medical prevention and treatment handbook from Zhejiang University School of Medicine[30] in 10 languages. Unfortunately the total amount of data from these sources do not exceed a few hundred sentences in each direction, so they are not enough for system development; they could, though, be potentially useful as an additional smaller evaluation set or for terminology extraction.

## 6  Baseline Results and Discussion

We present baseline results in some language directions, using the following systems:

1. For English to es, fr, pt, ru, sw, id, ln, lg, mr and most opposite directions: we use the OPUS-MT systems (Tiedemann and Thottingal, 2020) which are trained on the OPUS parallel data (Tiedemann, 2012a) using the Marian toolkit (Junczys-Dowmunt et al., 2018). We also use the pretrained systems between French and es, id, ln, lg, ru, rw.

2. For English to Russian we compare against the pre-trained Fairseq models that won the WMT Shared Task in this direction last year (Ng et al., 2019), as well as the English to French system of Ott et al. (2018). For translation between English and Chinese we also use a system trained on WMT'18 data (Bojar et al., 2018).

3. We train systems between English and ar, fa, mr, om, zu on publicly available corpora from OPUS (referred to as "our OPUS models").

4. We train multilingual systems between English and hi, ms, and ur on a TED talks dataset (Qi et al., 2018) ("our Multilingual TED models").

We note that none of the systems have been specifically trained or fine-tuned on any data listed above. We leave domain adaptation studies for future work.

**Results**  Table 3 in the Appendix §A presents results in translation from English to all languages whose MT systems we were able to train or use, while Table 4 includes results in the opposite directions. Similarly, Tables 5 and 6 in the Appendix §A show the quality of MT systems, as measured on our test set, from and to French for a few languages. All tables also include a breakdown of the quality for each test domain.[31]

**Discussion**  First and foremost, the main takeaway from these baseline results lie not in the above-mentioned Tables, but in the languages that are *not* present in them. We were unable to find either pre-trained MT systems or publicly available parallel data in order to train our own baselines for Dari, Pashto, Tigrinya, Nigerian Fulfulde, Kurdish Sorani, Myanmar, Oromo, Dinka, Nuer, and isiZulu.[32] This highlights the need for serious data collection efforts to expand the availability of data for large swathes of under-represented communities and languages.

Beyond this obvious limitation, the existing systems' results highlight the divide between high-resource language pairs and low-resource ones.

---

[23] https://www.bbc.com/
[24] https://www.voanews.com/
[25] https://www.dw.com/
[26] https://bit.ly/3iCeqnl
[27] https://welcoming.seattle.gov/covid-19/
[28] https://bit.ly/3e7Mq7M
[29] https://covid-no-mb.org/
[30] https://bit.ly/3gvxaTL

[31] Note, however, that each sub-domain posits a smaller test set than the complete set, and hence any result should properly take into account statistical significance measures.

[32] Note that although a small amount of parallel data exists for English-isiZulu, Abbott and Martinus (2019) report very low results on general benchmarks as the parallel data requires cleaning.

For all European languages (Spanish, French, Portuguese, Russian) as well as for Chinese and Indonesian, MT produces very competitive results with BLEU scores between 25 and 49.[33] In contrast, the output translations for languages like Lingala, Luganda, Marathi, or Urdu are quite disappointing, with extremely low BLEU scores under 10. The existence of pre-trained systems or of parallel data, hence, is not enough; this level of quality is basically unusable for any real-world deployment either for translators or for end-users.

A comparison of the results across different domains is also revealing. BLEU scores are generally higher on Wikipedia and news articles; this is unsurprising, as most MT systems rely on such domains for training, as they naturally produce parallel or quasi-parallel data. Our PubMed data pose a more challenging setting, but perhaps not as challenging as we initially expected, although the results vary across languages. In translating from English to French, for instance, the difference between Wikipedia and PubMed is more than 14 BLEU points,[34] while the differences are smaller for e.g. Indonesian-English (6 BLEU points) or Russian-English (4 BLEU points).

**Future Work**   Several concrete steps have the potential to improve MT for all languages in our benchmark. All results we report are with MT systems trained on general domain data or in particularly out-of-domain data (such as TED talks); domain adaptation techniques using small in-domain parallel resources or monolingual source- or target-side data should be able to increase performance. Incorporating the terminologies as part of the training and the inference schemes of the models could also ensure faithful and consistent translations of the COVID-19-specific scientific terms that might not naturally appear in other training data or might appear in different contexts.

Another direction for improvement involves multilingual NMT models trained on massive web-based corpora (Aharoni et al., 2019), which have improved translation accuracy particularly for languages in the lower end of data availability. Also viable are methods relying on multilingual model transfer, which can target languages with extremely small amounts of data, as in Chen et al. (2018).

Lastly, we need to improve the representation of low resource languages in public domain corpora. While there are open data collections in OPUS (Tiedemann, 2012b), and mined corpora like Paracrawl (Esplà et al., 2019), WikiMatrix (Schwenk et al., 2019), they don't cover enough low-resource languages. We hope that the availability of multilingual representations such as Multilingual BERT (Devlin et al., 2018) and XML-R (Conneau et al., 2019) will empower the creation of parallel corpora for low resource languages through low-resource corpus filtering (Koehn et al., 2019) or other approaches.

## 7   Conclusion

Enabling efficient and accurate communication through translations still has a ways to go for the majority of the world's languages and particularly the most vulnerable ones. With this effort we only address a fraction of the needs for a fraction of the world's languages. Nevertheless, we hope that the MT Resources that we release will have an immediate impact for the languages we cover. More importantly, the benchmark we release will allow the MT research community, both academic and industrial, to be more prepared for the next crisis where translation technologies will be needed.

## Acknowledgements

We would like to thank the people who made this effort possible: Tanya Badeka, Jen Wang, William Wong, Rebekkah Hogan, Cynthia Gao, Rachael Brunckhorst, Ian Hill, Bob Jung, Jason Smith, Susan Kim Chan, Romina Stella, Keith Stevens. We also extend our gratitude to the many translators and the quality reviewers whose hard work are represented in our benchmarks and in our translation memories. Some of the languages were very difficult to source, and the burden in these cases often fell to a very small number of translators. We thank you for the many hours you spent translating and, in many cases, re-translating content.

---

[33]Note that BLEU scores on test sets on different languages (e.g. in Tables 3 and 5) are not directly comparable.

[34]We note again that these scores are not directly comparable as the underlying test data are different.

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

## Appendix

## A    Baseline MT Results

## B    Source Documents

The list of original English-language documents that make up our benchmark are listed in Table 7.

## C    Quality Assurance Documents

The 558 sentences of our quality assurance set are comprised of examples from almost all subdomains of the corpus. Specifically, it includes 40 sentences from the conversational data, one PubMed document (PubMed_8), two of the Wikinews documents (Wikinews_1, Wikinews_3) and one complete Wikipedia article (Wikipedia_handpicked_4).

## D    Expected Quality per Language

For all translation directions the quality was quite good, with average quality scores above 95%. The detailed list of the quality evaluations on our sampled documents, which can be considered a proxy for the overall translation quality of our whole dataset, is available in Table 8.

## E    Terminology Terms

The **Facebook** terminologies provide translations from English into 129 languages/locales for the following terms: 1918 flu, acute bronchitis, acute respiratory disease, AIDS, airborne droplets, airway, alcohol-based, alcohol-based hand rub, alcohol-based hand sanitizer, alveolar disease, an epidemic, annual flu, anti-inflammatory drug, antiviral drug, antiviral treatment, assay, asymptomatic, avian flu, Black Plague, blood pressure, body fluid, breathing, bubonic plague, c19, cv19, case fatality rate, causative agent, CDC, chemical disinfection, chest x-ray, clinical diagnosis, clinical trial, close contact, cvirus, chronic respiratory disease, common cold, common flu, communicable disease, community spread, community transmission, compromised immune system, contagion, contagious, contagiousness, corona, corona virus, corona virus epidemic, corona virus outbreak, corona virus scare, coronavirus, coronavirus cases, coronavirus outbreak, coronavirus pandemic, coronavirus scare, cough, cough etiquette, coughing, cov 19, cov19, COVID, COVID-19 crisis, covid-19, covid19, COVID-19 epidemic, COVID-19 outbreak, COVID-19 pandemic, COVID 19, current crisis, current health crisis, current outbreak, current pandemic, CV, CV-19, deadly, deadly virus, death rate, death toll, decontaminate, detectable, detergent, difficulty breathing, diabetes, diagnosable disease, diagnostic protocol, diagnostic testing, disease itself, disease outbreak, disinfectant, disposable, droplet transmission, droplets, dry cough, dry surface contamination, ebola, effective treatment, electron microscope, emergency department, epidemic, epidemic curve, epidemic peak, epidemiologist, exposure, extreme caution, eye protection, face mask, face masks, family cluster, fatality rate, fecal contamination, fever, flatten the infection curve, flu, food safety, formaldehyde, gastroenteritis, germicide, global pandemic, global warming, good respiratory hygiene, Guangdong, H1N1, H1N1 virus, hand disinfectant, hand sanitizer, health care provider, health crisis, health plan carrier, health services, heart failure, high fever, HIV, hospitalize, household bleach, Hubei, hydrogen peroxide, hygiene, immune system, immunity, immunocompromised, immunologist, incubation, incubation period, indirect contact, infect, infection, infectious, infectivity, influenza, initial transmission event, intensive care equipment, intermediate host, intrauterine, intubation, isolation, Isopropanol, laboratory test kit, laboratory testing, lack of testing, lack of tests, latest updates, liver failure, local public health authority, lockdown, lungs, masks, measles, mechanical ventilation, media coverage, medical care, MERS, MERS CoV, microbiologist, mild cough, mortality rate, multi-organ failure, nasal congestion, neutral detergent, new normal, novel corona virus, novel coronavirus, novel coronavirus outbreak, novel virus, online medical consultation, onset, outbreak, outbreak readiness, overall case fatality, pandemic, pangolin, pathogen, pathology, patient care equipment, personal protective equipment, person-to-person transmission, phlegm, physical contact, plague, pneumonia, pneumonias, positive test, precaution, precautionary, pre-existing condition, preparedness, prognosis, protect myself, protect others, protect yourself, protective measures, public health crisis,

pulmonary tissue damage, quarantine, rapid risk assessment, reagent, regular flu, reinfection, renal disease, renal failure, respirator, respiratory, respiratory disease, respiratory distress, respiratory droplets, respiratory hygiene, respiratory illness, respiratory syndrome, respiratory tract disease, restrictions, resurge, RNA, rubeola, runny nose, SARS, SARS-CoV-2, SARS-CoV, sars-related coronavirus, seasonal flu, seasonal influenza, secretion, self-quarantine, sepsis, septic shock, Severe Acute Respiratory Syndrome, shortness of breath, sickened, sickness, sneeze, sneezes, sneezing, social distancing, sore throat, spanish flu, specimen collection, spread, sputum, stigmatisation, supportive, surface, surfaces, suspected infection, swab, swine flu, symptom, symptomatic, symptoms, targeted disinfection, test kit, testing error, tissue damage, tp shortage, toilet paper shortage, touch, touching, transmissibility, transmissible, transmission, transmission potential, typical flu, underlying disease, updates, vaccine, vaccines, ventilation, ventilator, ventilators, viral infection, viral outbreak, viral pandemic, viral pneumonia, virtual care, virulence, virus itself, virus outbreak, virus scare, virus spreads, virus strain, viral transmission, vivid-19, washing hands, washing ones hands, wash your hands, white blood cell count, widespread transmission, working from home, WFH, Wuhan, aetiology, alpha coronavirus, alveolus, antibody test, antibody therapy, antigen, antimicrobial agent, antiviral antibody, associated disease, betacoronavirus, bioaerosol, bronchoalveolar, canine coronavirus, cDNA, complete genome, confirmatory testing, coronavirus envelope, cytological, diabetes mellitus, diclofenac, dyspnoea, ectodomain, emergency airway management, envelope antibody, fomite, food safety, genome analysis, glycoprotein, gp surgery, hemoptysis, histological examination, host cell, Immunofluorescence, immunopathogenesis, interferon, Lancet, lopinavir, microbial flora, nucleic acid test, pan-coronavirus assay, passive antibody therapy, pathogenesis, pathophysiology, phylogenetic analysis, phylogenic tree, PPE, prophylaxis, protein sequence analysis, reverse transcription polymerase chain reaction, serological test, sodium hypochlorite, thermal disinfection, ultraviolet light exposure, vertical transmission, viral antigen, viral transmission, virology, virucidal.

The **Google** terminologies provide translations from English into 100 languages/locales of various amounts of terms. There are 27 more translation directions provided, for which we don't provide details for lack of space. The following subset of terms, though, should be common across all English-to-X terminologies: 14 days in isolation, 14 days quarantine, 2019 coronavirus, 2019 novel coronavirus, 2019-nCoV, 2020 coronavirus, 2020 novel coronavirus, about coronavirus, acute respiratory distress syndrome, advanced hand sanitizer, Affected by coronavirus, after exposure, After the epidemic, After the outbreak, airborne virus, alcohol based hand sanitizer, alcohol hand sanitizer, alcohol-based hand sanitizer, Anti-Corona virus spray, antibacterial hand sanitizer, ARDS, avoid exposure, be infected, Beware of coronavirus, bilateral interstitial pneumonia, CDC, Center for Disease Control, community spread, compulsory quarantine, contagious, coronavirus, coronavirus (COVID-19), coronavirus alert, coronavirus cases, coronavirus concerns, coronavirus crisis, coronavirus disease, Coronavirus disease (COVID-19) outbreak, coronavirus early symptoms, coronavirus epidemic, coronavirus exposure, coronavirus incubation, coronavirus incubation period, coronavirus infection, coronavirus map, coronavirus medicine, coronavirus medicines, coronavirus news, coronavirus outbreak, coronavirus pandemic, coronavirus pneumonia, coronavirus precautions, coronavirus prevention, coronavirus protection, coronavirus quarantine, coronavirus SOS Alert, coronavirus spread, coronavirus symptoms, coronavirus transmission, coronavirus travel ban, coronavirus travel restrictions, coronavirus treatment, coronavirus update, coronavirus vaccine, coronavirus vaccines, covid cases, covid early symptoms, covid incubation period, covid international spread, covid international travel, covid isolation, covid map, covid medicine, covid medicines, covid news, covid outbreak, covid pandemic, covid panic, covid SOS Alert, covid symptoms, covid transmission, covid travel ban, covid travel restrictions, covid treatment, covid vaccine, covid vaccines, covid-19, covid-19 alert, covid-19 cases, covid-19 CDC, covid-19 contagious, covid-19 cure, covid-19 dangerous, covid-19 deadly, covid-19 death, covid-19 deaths, covid-19 domestic travel, covid-19 early symptoms, covid-19 effects, covid-19 epidemic, covid-19 exposure, covid-19 fatal, covid-19 fever, covid-19 illness, covid-19 incubation, covid-19 incubation period, covid-19 infection, covid-19 international spread, covid-19 international travel, covid-19 isolation, covid-19 lockdown, covid-19 map, covid-19 medicine, covid-19 medicines, covid-19 news, covid-19 outbreak, covid-19 pandemic, covid-19 panic, covid-19 precautions, covid-19 protection, covid-19 quarantine, covid-19 SOS Alert, covid-19

spread, covid-19 symptoms, covid-19 transmission, covid-19 travel ban, covid-19 travel restrictions, covid-19 treatment, covid-19 uncontrolled spread, covid-19 vaccine, covid-19 vaccines, COVID-19 virus, covid-19 virus outbreak, covid-19 virus transmission, covid-19 WHO, covid19, covid19 alert, covid19 cases, covid19 CDC, covid19 deaths, covid19 domestic travel, covid19 effects, covid19 epidemic, covid19 exposure, covid19 fatal, covid19 fever, covid19 illness, covid19 incubation, covid19 incubation period, covid19 infection, covid19 international spread, covid19 international travel, covid19 isolation, covid19 lockdown, covid19 map, covid19 medicine, covid19 medicines, covid19 news, covid19 outbreak, covid19 pandemic, covid19 precautions, covid19 protection, covid19 quarantine, covid19 SOS Alert, covid19 spread, covid19 symptoms, covid19 transmission, covid19 travel ban, covid19 travel restrictions, covid19 treatment, covid19 vaccine, covid19 vaccines, covid19 virus, covid19 virus outbreak, covid19 virus transmission, current outbreak, deadly outbreak, disease outbreak, Disposable hand sanitizer, domestic travel, droplets, Effects of coronavirus, epidemic, epidemic and pandemic, epidemic disease, epidemic outbreak, epidemic period, epidemic prevention, epidemic season, epidemic situation, exposure, exposure time, fever, fight the virus, Fighting the outbreak, flu epidemic, fomites, global health emergency, global outbreak, global pandemic, hand sanitizer, hand sanitizer dispenser, hand sanitizer gel, hand sanitizer spray, home isolation, home quarantine, illness, incubation period, infected, instant hand sanitizer, international spread, international travel, isolation, isolation period, isolation room, isolation valve, isolation ward, lockdown, major outbreak, mandatory quarantine, mass gathering, medicine, medicines, n95, n95 mask, n95 respirator, ncov, ncov-2019, new coronavirus, new coronavirus pneumonia, novel coronavirus, novel coronavirus infection, novel coronavirus outbreak, novel coronavirus pneumonia, ongoing outbreak, outbreak, outbreak of coronavirus, outbreak of disease, pandemic, pandemic influenza, pandemic outbreak, pandemic plan, pandemic potential, pneumonia, pneumonia epidemic, potential exposure, precautions, prevent virus, prolonged exposure, quarantine, quarantine area, quarantine facility, quarantine measures, quarantine period, quarantine room, quarantine zone, Recovered coronavirus patient, repatriate, repeated exposure, respiratory syncytial virus, respiratory virus, Sanitizing hand sanitizer, SARS-CoV-2, self isolation, self quarantine, Severe outbreak, social distancing, social distancing measures, social isolation, SOS Alert, spread, spread of coronavirus, spread of virus, strain of virus, the incubation period, the novel coronavirus, touching face, travel advisory, travel ban, travel restrictions, use hand sanitizer, vaccines, viral outbreak, virtual lockdown, virus, virus carrier, virus infection, virus mask, virus outbreak, virus prevention, virus protection, virus spread, virus spreads, virus strain, virus transmission, wash your hands, washing hands, WHO, WHO Confirmed, WHO Deaths, widespread outbreak, World Health Organization, zoonotic disease, zoonotic virus.

| en→ | Translation Accuracy by Domain (BLEU) | | | | | |
|---|---|---|---|---|---|---|
| | *Overall* | PubMed | Conv. | Wikisource | Wikinews | Wikipedia |
| ***HelsinkiNLP OPUS-MT*** | | | | | | |
| es-LA | 48.73 | 49.87 | 32.11 | 39.73 | 53.20 | 48.70 |
| es-LA[†] | *49.25* | *50.17* | *30.60* | *40.74* | *53.29* | *49.33* |
| fr | 37.59 | 27.11 | 30.86 | 39.72 | 28.44 | 42.69 |
| id | 41.27 | 37.75 | 28.68 | 40.52 | 42.85 | 43.2 |
| pt-BR | 47.26 | 46.64 | 30.85 | 36.52 | 48.21 | 48.32 |
| pt-BR[†] | *47.27* | *47.63* | *24.90* | *36.11* | *49.26* | *47.94* |
| ru | 25.49 | 21.65 | 18.43 | 18.40 | 24.24 | 27.84 |
| sw | 22.62 | 19.94 | 19.59 | 27.52 | 26.79 | 23.61 |
| lg | 2.96 | 2.54 | 1.71 | 5.17 | 3.37 | 3.01 |
| ln | 7.85 | 8.33 | 5.40 | 12.0 | 7.42 | 7.38 |
| mr | 0.21 | 0.18 | 0.96 | 0.30 | 0.62 | 0.19 |
| ***Fairseq*** | | | | | | |
| fr | 36.96 | 26.83 | 27.71 | 41.70 | 27.55 | 42.35 |
| ru | 28.88 | 26.45 | 15.89 | 21.52 | 28.95 | 30.61 |
| ***WMT-18*** | | | | | | |
| zh | 33.70 | 41.66 | 16.26 | 23.35 | 28.51 | 30.10 |
| ***Our OPUS models*** | | | | | | |
| ar | 15.16 | 11.81 | 10.47 | 11.38 | 14.36 | 17.50 |
| fa | 8.48 | 7.85 | 3.88 | 14.54 | 5.26 | 8.72 |
| prs♣ | 9.49 | 7.73 | 2.39 | 10.40 | 9.08 | 10.51 |
| mr | 0.12 | 0.09 | 1.07 | 0.24 | 0.30 | 0.09 |
| om | 0.57 | 0.53 | 0.75 | 1.25 | 0.47 | 0.54 |
| zu | 11.73 | 13.98 | 16.42 | 16.74 | 14.12 | 10.18 |
| ***Our Multilingual TED*** | | | | | | |
| hi | 6.43 | 5.45 | 10.37 | 6.34 | 3.08 | 6.80 |
| ms | 6.26 | 5.90 | 6.12 | 10.15 | 6.17 | 6.17 |
| id | 25.65 | 23.76 | 20.31 | 27.92 | 24.40 | 26.54 |
| ur | 2.79 | 2.79 | 6.48 | 5.02 | 4.0 | 2.40 |
| ***WMT-20*** | | | | | | |
| zh | 57.83 | 68.88 | 41.49 | 33.57 | 55.97 | 53.45 |
| ps | 36.56 | 49.26 | 26.94 | 12.15 | 8.85 | 32.25 |
| ru | 40.20 | 29.71 | 26.37 | 22.90 | 40.44 | 46.38 |

Table 3: Baseline results on some English-to-X translation directions. †: the *italicized* rows of the Spanish and Portuguese results show the quality of the outputs obtained using the *European* Spanish and Portuguese systems, while the top lines use es_MX and pt_BR tags. ♣: the results on Dari (prs) are with the English-Farsi (fa) model.

| ☼→en | | Translation Accuracy by Domain (BLEU) | | | | |
| --- | --- | --- | --- | --- | --- | --- |
| | *Overall* | PubMed | Conv. | Wikisource | Wikinews | Wikipedia |
| ***Naver Papago*** | | | | | | |
| es-LA | 52.55 | 54.31 | 35.47 | 45.60 | 50.25 | 52.50 |
| es-LA$^\diamond$ | 52.78 | 54.16 | 34.82 | 44.90 | 51.24 | 52.97 |
| fr | 41.65 | 30.58 | 32.85 | 43.57 | 27.07 | 47.95 |
| fr$^\diamond$ | 42.12 | 30.84 | 32.17 | 43.82 | 28.67 | 48.61 |
| ***HelsinkiNLP OPUS-MT*** | | | | | | |
| es-LA | 46.82 | 49.23 | 33.02 | 39.99 | 44.69 | 46.38 |
| fr | 39.40 | 28.78 | 26.10 | 39.55 | 28.10 | 45.09 |
| id | 34.86 | 33.40 | 26.52 | 38.64 | 37.79 | 35.35 |
| pt-BR | 48.56 | 49.62 | 31.52 | 40.05 | 43.11 | 40.19 |
| ru | 28.53 | 26.43 | 21.70 | 27.27 | 25.39 | 29.94 |
| hi | 18.91 | 16.99 | 23.60 | 22.22 | 14.68 | 19.71 |
| rw | 8.29 | 7.67 | 7.56 | 13.50 | 7.53 | 8.31 |
| lg | 5.62 | 4.66 | 5.54 | 7.44 | 5.31 | 6.04 |
| ln | 6.71 | 7.03 | 1.86 | 9.97 | 4.58 | 6.41 |
| ***WMT-18*** | | | | | | |
| zh | 28.94 | 32.61 | 16.82 | 17.11 | 31.76 | 27.68 |
| ***Our OPUS models*** | | | | | | |
| ar | 28.56 | 25.25 | 13.39 | 23.07 | 23.82 | 31.22 |
| fa | 15.07 | 12.00 | 23.44 | 23.68 | 14.78 | 16.42 |
| prs$^\clubsuit$ | 15.16 | 15.19 | 15.76 | 20.02 | 15.57 | 14.72 |
| mr | 1.16 | 1.02 | 1.46 | 1.74 | 1.80 | 1.56 |
| om | 2.11 | 1.72 | 1.90 | 4.41 | 2.87 | 2.14 |
| zu | 25.52 | 26.32 | 22.25 | 30.03 | 28.03 | 24.75 |

Table 4: Baseline results on some X-to-English translation directions. ♣: the results on Dari (prs) are with the Farsi (fa)–English model. ◇: results with the systems adapted to the medical domain.

| fr→☼ | | Translation Accuracy by Domain (BLEU) | | | | |
| --- | --- | --- | --- | --- | --- | --- |
| | *Overall* | PubMed | Conv. | Wikisource | Wikinews | Wikipedia |
| ***HelsinkiNLP OPUS-MT*** | | | | | | |
| en | 39.40 | 28.78 | 26.10 | 39.55 | 28.10 | 45.09 |
| es-LA | 34.95 | 26.28 | 25.31 | 31.42 | 29.25 | 39.87 |
| ru | 15.11 | 10.49 | 13.66 | 16.73 | 10.38 | 17.50 |
| rw | 3.83 | 2.40 | 3.35 | 3.66 | 2.30 | 4.56 |
| lg | 1.48 | 1.05 | 1.34 | 1.73 | 0.77 | 1.69 |
| ln | 6.14 | 5.25 | 3.59 | 9.71 | 2.16 | 6.61 |

Table 5: Baseline results on some French-to-X translation directions.

| ○→**fr** | | **Translation Accuracy by Domain (BLEU)** | | | | |
|---|---|---|---|---|---|---|
| | *Overall* | PubMed | Conv. | Wikisource | Wikinews | Wikipedia |
| *HelsinkiNLP OPUS-MT* | | | | | | |
| es-LA | 29.21 | 22.95 | 22.24 | 29.88 | 21.95 | 32.63 |
| en | 37.59 | 27.11 | 30.86 | 39.72 | 28.44 | 42.69 |
| id | 18.95 | 13.59 | 17.92 | 22.83 | 14.02 | 21.48 |
| ru | 17.62 | 12.94 | 18.34 | 19.82 | 13.52 | 19.96 |
| lg | 2.91 | 1.76 | 1.81 | 5.30 | 0.81 | 3.32 |
| ln | 4.77 | 3.62 | 2.88 | 7.74 | 2.96 | 5.22 |
| rw | 5.62 | 3.96 | 2.18 | 8.18 | 2.89 | 6.39 |

Table 6: Baseline results on some X-to-French translation directions.

| Doc ID | Type | Source/URL |
|---|---|---|
| *Conversational* | | |
| CMU_1 | medical-domain phrases | http://www.speech.cs.cmu.edu/haitian/text/1600_medical_domain_sentences.en |
| *PubMed* | | |
| PubMed_6 | Scientific Article | https://www.ncbi.nlm.nih.gov/pmc/articles/PMC7096777/ |
| PubMed_7 | Scientific Article | https://www.ncbi.nlm.nih.gov/pmc/articles/PMC7098028/ |
| PubMed_8 | Scientific Article | https://www.ncbi.nlm.nih.gov/pmc/articles/PMC7098031/ |
| PubMed_9 | Scientific Article | https://www.ncbi.nlm.nih.gov/pmc/articles/PMC7119513/ |
| PubMed_10 | Scientific Article | https://www.ncbi.nlm.nih.gov/pmc/articles/PMC7124955/ |
| PubMed_11 | Scientific Article | https://www.ncbi.nlm.nih.gov/pmc/articles/PMC7125052/ |
| *Wikipedia* | | |
| Wikipedia_hand_1 | Wikipedia Article | https://en.wikipedia.org/wiki/2019%E2%80%9320_coronavirus_pandemic |
| Wikipedia_hand_3 | Wikipedia Article | https://en.wikipedia.org/wiki/COVID-19_testing |
| Wikipedia_hand_4 | Wikipedia Article | https://en.wikipedia.org/wiki/Hand_washing |
| Wikipedia_hand_5 | Wikipedia Article | https://en.wikipedia.org/wiki/Impact_of_the_2019%E2%80%9320_coronavirus_pandemic_on_education |
| Wikipedia_hand_7 | Wikipedia Article | https://en.wikipedia.org/wiki/Workplace_hazard_controls_for_COVID-19 |
| Wiki_20 | Wikipedia Article | https://en.wikipedia.org/wiki/Angiotensin-converting_enzyme_2 |
| Wiki_26 | Wikipedia Article | https://en.wikipedia.org/wiki/Bat_SARS-like_coronavirus_WIV1 |
| Wiki_7 | Wikipedia Article | https://en.wikipedia.org/wiki/COVID-19_apps |
| Wiki_13 | Wikipedia Article | https://en.wikipedia.org/wiki/COVID-19_drug_development |
| Wiki_10 | Wikipedia Article | https://en.wikipedia.org/wiki/COVID-19_drug_repurposing_research |
| Wiki_29 | Wikipedia Article | https://en.wikipedia.org/wiki/COVID-19_in_pregnancy |
| Wiki_11 | Wikipedia Article | https://en.wikipedia.org/wiki/COVID-19_surveillance |
| Wiki_4 | Wikipedia Article | https://en.wikipedia.org/wiki/COVID-19_vaccine |
| Wiki_5 | Wikipedia Article | https://en.wikipedia.org/wiki/Coronavirus |
| Wiki_9 | Wikipedia Article | https://en.wikipedia.org/wiki/Coronavirus_disease_2019 |
| *Wikinews* | | |
| Wikinews_1 | News Segment | https://en.wikinews.org/wiki/Bangladesh_reports_five_new_deaths_due_to_COVID-19,_a_daily_highest |
| Wikinews_2 | News Segment | https://en.wikinews.org/wiki/National_Basketball_Association_suspends_season_due_to_COVID-19_concerns |
| Wikinews_3 | News Segment | https://en.wikinews.org/wiki/SARS-CoV-2_surpasses_one_million_infections_worldwide |
| Wikinews_4 | News Segment | https://en.wikinews.org/wiki/Stores_in_Australia_lower_toilet_paper_limits_per_transaction |
| Wikinews_5 | News Segment | https://en.wikinews.org/wiki/US_President_Trump_declares_COVID-19_national_emergency |
| Wikinews_6 | News Segment | https://en.wikinews.org/wiki/World_Health_Organization_declares_COVID-19_pandemic |
| *Wikivoyage* | | |
| Wikivoyage_1 | Public Service Announcement | https://en.wikivoyage.org/wiki/2019%E2%80%932020_coronavirus_pandemic |
| *Wikisource* | | |
| Wikisource_1 | Executive Order | https://en.wikisource.org/wiki/California_Executive_Order_N-33-20 |
| Wikisource_1 | Communiqué | https://en.wikisource.org/wiki/Covid-19:_Lightening_the_load_and_preparing_for_the_future |

Table 7: List of all source documents for our translation benchmark.

| en→☐ | QA score (%) (initial (→) final | re-worked | en→☐ | QA score (%) (initial →) final | re-worked |
|---|---|---|---|---|---|
| ar | 99.28 | | ha | 94.29 | ✓ |
| zh | 98.89 | | km | 89.15→91.23 | ✓ |
| fr | 95.78 | ✓ | am | 84 → 93.85 | ✓ |
| pt-BR | 97.85 | ✓ | zu | 97.77 | ✓ |
| es-419 | 99.60 | | tl | 98.08 | ✓ |
| hi | 98.39 | ✓ | ms | 98.32 | |
| ru | 98.36 | | ta | 79 → 94.74 | ✓ |
| sw | 95.43 | ✓ | my | 97.68 | |
| id | 97.98 | | bn | 94.78 | ✓ |
| ku | 99.96 | | ur | 94 | ✓ |
| ln | 96.13 | | fa | 95.37 → 96.35 | ✓ |
| lg | 99.63 | | mr | 92 → 96.45 | ✓ |
| ne | 95.32 | ✓ | om | 94.25 | ✓ |
| pus | 98.96 | | ckb | 99.94 | |
| ti-ET | 81.98 → 93.47 | ✓ | din | 99.40 | |
| so | 94.80 | ✓ | kr | 98.88 | |
| prs | 96.34 | ✓ | ti-ER | 90.98 | ✓ |
| fuv-Latn-NG | 99.41 | | swc | | |
| rw | 99.73 | | nus | | |
| Average QA score (final): 96.77 | | | | | |

Table 8: QA score across languages. We report the initial QA score and whether re-work was required on the produced translations (beyond the QA sample, due to serious translation errors). In cases where the initial QA yielded very poor results, the translations were corrected in their entirety and a new QA process was performed, and we report both the initial and the final QA results. Note: a "N/A" final QA score in the score indicates that the final score is temporarily not available and we will report it in an updated version of the paper.