# OpenReview forum: "TICO-19: the Translation Initiative for COvid-19"
_EMNLP/2020/Workshop/NLP-COVID — NLP-COVID19-EMNLP Oral_

### Official Review · AnonReviewer2 · 2020-09-21
**Great resource**

**Rating:** 9
**Confidence:** 4

**Review:**

The authors describe the TICO dataset, a development and testing dataset for multi-lingual medical translation. They describe in great detail the creation of this dataset, and the QA measures they undertook. I am particularly a fan of Table 8 which includes specific measures and details about which dataset underwent which parts of their QA process, although I note there appears to be an extra empty aggregate row.

The authors provide a great overview of the varying languages, grouping them for relative importance in the translation effort, and making a qualitative assessment of the resources available for each of them.

Overall I think that this is a high quality development and testing set. I would be happy to see this paper accepted, although I think there are some easily tackleable areas for improvement. My concerns are below:

It's not clear to me whether or the not the authors intend on releasing their systems (licensing of the data?). As is, I would assume not, which would be unfortunate but not a reason to stop publication.

I disagree with the relative downweighting of the importance of the parallel data in the paper - the authors should document the sizes of the data they found, even if small.  Licensing concerns are relevant for others using the data, but a proper description of it, including sizes and genres, belongs in this paper.

I am concerned that the relative importance of the PubMed data in the evaluation datasets, by a small number of documents providing a large number of sentences, will primarily reward systems for a somewhat random game of vocabulary whack-a-mole instead of rewarding the ability to produce a high quality translation over a diverse set of inputs. While adjusting this is likely impossible at this point due to expense or annotation effort, I would like to see an explicit treatment of this potential issue.

Wherever possible I would request that the authors create wayback or archived versions of the links in the paper and to prefer full links over shortened ones instead of relying on bit.ly's promise to be around forever. This will help ensure that their work remains accessible from a long term perspective.

---

> ### Author Response · Authors · 2020-09-26
> **Response to Reviewer 2**
>
> Thank you for your review and thorough comments!
> Let us address your questions here:
>
> 1. Empty aggregate row in Table 8: at the time of submission, the QA process for the data in the Ethiopian languages of TICO-19 (namely, Amharic and Tigrinya) had not been finished, due to delays caused by the internet-shutdown-situation in Ethiopia, where the evaluators reside. As of last week, the QA process has been completed, and we will be updating the paper and the dataset as soon as possible (and certainly before the camera-ready submission).
>
> 2. The MT models that we have used to provide baseline results are either already available (the OPUS systems are available here: https://huggingface.co/Helsinki-NLP) or trained on publicly available data (most can be collectively found here: http://opus.nlpl.eu/). We can certainly plan to release these models, as there's no licensing issues involved. We also want to point out that the data that we created (the TICO-19 evaluation benchmark) is available under a CC-0 license.
>
> 3. We appreciate your point that we should report the statistics of any available monolingual/parallel data. We will have to confirm this before the camera-ready, but licensing issues should *probably* not prohibit us from describing the data. If this is the case, we will update the paper with another Table in the Appendix, denoting, as you suggest, sizes and genres.
>
> 4. This is a good point, although at this point there really is not much we can do to address this potential issue. We did consciously include as much PubMed data as possible, as we consider them critical. Plus, scientific medical translation benchmarks for low-resource languages are severely lacking, as most research does not typically go beyond the EMEA corpus and other smaller collections that were used in e.g. the WMT Medical translation shared task http://statmt.org/wmt14/medical-task/index.html, which are severely European-centric). As we begin to evaluate the MT systems more thoroughly, we (and other researchers) will be able to investigate the extend to which the importance of only having a few long documents in the PubMed data is an issue or not. If this does prove to be a problem, we could investigate whether a different dev/test split could perhaps alleviate it, or if more documents will need to be translated and included in the benchmark.
>
> 5. Great point regarding wayback links. We will certainly make sure to update all links in the camera-ready!

---

### Official Review · AnonReviewer3 · 2020-09-23
**TICO-19: the Translation Initiative for COvid-19**

**Rating:** 5
**Confidence:** 4

**Review:**

**Summary**
This work described a translation system TICO-19 based on multiple sources including English PubMed and Wikipedia. English documents were translated into 35 languages with terminologies provided by Facebook and Google, where the OPUS-MT system was employed.

**Comments**
1. In the Quality Assurance subsection, It is not clear how the first editing is performed? How do you solve the disagreements? Can you provide examples?

2. In the second round QA process, how the 95% rate was achieved? How do you evaluate?

3. The MT is pre-trained on OPUS parallel data. It is not clear how do you use the resources in section 5 to adjust the pre-trained system.

4. Can you please clarify the structure of the pipeline system? After collecting the data and specifying the target language, how do Facebook, Google terminology datasets being used in the MT system?

---

> ### Author Response · Authors · 2020-09-26
> **Response to Reviewer 3**
>
> Thank you for your review!
> Let us address your questions and comments:
>
> 1. Quality Assurance: The first editing pass is handled by the Language Service Providers (LSPs). Our instructions to the LSPs explicitly required that the first-pass translation undergoes a round of editing by a qualified translator different than the one who actually performed the translation. This is standard practice in the translation industry when high-quality translations are requested. The LSPs handle the process (and the actual way to handle disagreements can vary among them) but the process involves both the editor and the translator resolving the disagreements.
> 2. Second-Pass QA: the process for the second-pass quality assurance involves trained translators with the necessary in-domain expertise, who annotate the translations if they find any errors and create corrections wherever needed. All errors are annotated with a label regarding error type (Grammar, Punctuation, Spelling, Capitalization, Addition/Omission, Mistranslation, unnaturalness, untranslated text) and of severity (minor, major, critical). The final score (with a target of above 95% in the final samples) is computed by aggregating these annotations (due to our partners being commercial, the exact formula for calculating the final score cannot be released). We will, however, release both the initial and the corrected sentences, along with the error labels.
> 3. You are right that the MT systems we use to provide baseline results are only trained on general-domain data. We did not perform and fine-tuning or domain adaptation using the resources from Section 5. We will update the paper to make this more clear.
> 4. We are a bit unclear on what you refer to as the "pipeline" system. We clarify again that we only provide baseline results with the pre-trained systems. The MT systems, currently, do not make use of the terminologies that we have created (we will change the wording in the paper to clarify this). We do, however, plan to investigate ways to incorporate them into the system, either during training, or during fine-tuning, or during domain adaptation, or even during inference.

---

### Official Review · AnonReviewer1 · 2020-09-25
**Great initiative that meets the purpose of the workshop**

**Rating:** 8
**Confidence:** 4

**Review:**

The paper asserts the importance of translation technologies in crisis situations and describes the TICO-19 benchmark for validating and testing translation systems for low-resourced languages. Also, other additional resources such as translation memories and in-domain monolingual data are presented. The paper concludes with initial benchmarked results using publicly available pre-trained models.

I think the work is well-aligned with the workshop's objective. I would like to applaud the authors for their efforts to construct such a benchmark with the under-served languages in mind.

I do have a few questions:

1. What does the 95% rate correspond to? Does it correspond to the direct assessment scheme employed in WMT?
Were there any domain-specific evaluation guidelines?

2. If I understood the paper correctly, the additional resources in Section 5 are not used at all to train/fine-tune the "Our OPUS" and "Our TED" models? Is this correct?

---

> ### Author Response · Authors · 2020-09-26
> **Response to Review 1**
>
> Thank you for your review!
> To answer your questions:
> 1. Quality Assurance score (95%): the process for the second-pass quality assurance involves a trained translator with the necessary in-domain expertise, who annotate the translations with errors and create corrections wherever needed. All errors are annotated with a label regarding error type (Grammar, Punctuation, Spelling, Capitalization, Addition/Omission, Mistranslation, unnaturalness, untranslated text) and of severity (minor, major, critical). The final score (with a target of above 95% in the final samples) is computed by aggregating these annotations (due to our partners being commercial, the exact formula for calculating the final score cannot be released). We will, however, release both the initial and the corrected sentences, along with the error labels.
> 2. You are correct that the additional resources in Section 5 are not used in our experiments. The OPUS and TED models are all trained on general-domain data. We are indeed starting to explore domain adaptation techniques that can utilize the additional resources in Section 5, but we are still at a preliminary stage.